# Determinants of Continuous Usage Intention in Community Group Buying Platform in China: Based on the Information System Success Model and the Expanded Technology Acceptance Model

**DOI:** 10.3390/bs13110941

**Published:** 2023-11-17

**Authors:** Yingjie Song, Lin Gui, Hong Wang, Yanru Yang

**Affiliations:** School of Economics and Management, Beijing University of Agriculture, Beijing 102206, China; 202130412008@bua.edu.cn (Y.S.); wanghongbit@126.com (H.W.); yyr2023@163.com (Y.Y.)

**Keywords:** community group buying, technology acceptance theory, information system success theory, continuous usage intention

## Abstract

Community group buying is a new retail model with broad development prospects. The community group buying model in China has brought obvious social and economic benefits. However, the continuous usage rate on some community group buying platforms is relatively low. Consumers’ continuous usage intentions are closely related to the sustainable development of community group buying platforms. Therefore, this study integrates the information system success model (D&M model) and the expanded technology acceptance model (TAM) to construct a research model that explores the factors influencing consumers’ continuous usage intentions from both the platform’s and consumers’ perspectives. The survey data involving 418 respondents who used community group buying platforms were developed and analyzed for structural equation model (SEM) testing. The results show the following: (1) Perceived usefulness, perceived ease of use, service quality, perceived trust, expectation confirmation, and subjective norms significantly affect continuous usage intention. (2) Subjective norms are significantly related to continuous usage intention. Perceived usefulness, perceived ease of use, service quality, perceived trust, and expectation confirmation indirectly affect continuous usage intention through user satisfaction. This research provides a new perspective for the theoretical research of community group buying and helps to promote the sustainable development of community group buying platforms in management practice.

## 1. Introduction

With the rapid development of e-commerce, the operation model of community group buying has seen significant market potential and offers vast market potential. The scale of community group buying has expanded from CNY 8.5 billion in 2018 to approximately CNY 300 billion in 2021, which is sufficient to generate the next growth curve of Internet giants. Community group buying is a new business model and consumption trend. Looking at the future development trend, consumer behavior will evolve towards personalization and e-commerce. Online group buying is a social or collective online shopping community in which a product or service item can be purchased at a significantly reduced price based on the critical quality of the purchase. Online group buying has two characteristics: the purchase price is lower than the market price and the items are sold in large quantities [1,2]. Community group buying builds upon online group buying by integrating an offline component for easier product receipt. Social services have always aimed to integrate the users they serve directly into the wider community. The United States is one of the earliest countries to use the community group buying model. For example, Groupon in the United States displays products that can be bought in a group every day and offers discounts of varying degrees for these products [3,4]. This model has developed rapidly in China, with continuous innovation and iteration. Currently, many e-commerce enterprises are learning from China’s community group buying model, aiming to tap into the market and have the same group buying capability as the Chinese market [5]. However, in the past two years, several community group buying platforms have folded due to improper operation.

For the community group buying model, the way it makes its users have the characteristics of high frequency and high stickiness purchase determines the future development prospects and profitability of the community group buying platform. Users of community retail stores experience a sense of community while shopping, and consumers perceive a sense of belonging to other shoppers by sharing store activities, similarities with other shoppers, and shopping preferences [6]. From a customer acquisition perspective, users serve as the traffic for the group buying platform. To achieve user retention, it is necessary to use big data to grasp the user portrait of using community group buying, accurately divide the marketing market, and optimize marketing strategies [7].

Previous studies have explored the influencing factors of using online platforms, but with limited focus on continuous usage. Perceived benefit is the key. The benefit is the value that customers expect in the consumption process, and the perceived benefit (PB) is the benefit that consumers can perceive a specific product. The factors affecting the willingness to participate in online group buying include consumer trust, satisfaction, reciprocity, perceived behavior, peer norms, service quality, organizational structure, organizational system, and reputation, among others. Trust is the most influential factor [8,9,10]. Although consumers’ acceptance of community group buying platforms is relatively high, it is also more likely to expose the gap between expectations and natural experience.

In conclusion, previous studies have mainly focused on aspects such as the definition of community group buying, consumer characteristics, and factors influencing consumer usage, with a lack of research on continuous usage. However, the community group buying industry is experiencing constant evolution, and significant changes have taken place in their market characteristics and consumer psychology. The community group buying platform has encountered problems such as intense competition and homogenization [4,11]. By the end of 2022, more than six community group buying platforms in China closed down. These closed-down platforms all have a common feature: low user stickiness. Resolving this issue helps to promote the sustainable development of community group buying platforms. Therefore, this article focuses on the community group buying sector and explores the influencing mechanisms of consumers’ continuous usage intentions in community group buying platforms after 2022. To a certain extent, this article updates the timeliness of research related to the community group purchase industry and reflects the latest usage characteristics of consumers. This article integrates the information system success model and the expanded technology acceptance model to create a new comprehensive theoretical model from multiple perspectives. The model combines variables from both the platform and consumer aspects, enabling a more comprehensive and accurate assessment of the factors that influence consumers’ continuous usage intentions. The purpose of this article is to explore the decisive factors affecting consumers’ preferences to continue using community group purchase platforms in order to provide practical enlightenment for the sustainable development of these platforms.

Therefore, this paper is organized as follows: Section 2 briefly reviews the literature on the D&M model and TAM. Section 3 comprises the research hypothesis development. Section 4 explains the methodology. The data analysis is presented in Section 5. The discussion and implications are presented in Section 6. Section 7 points out the limitations of this study and the directions for future research. The conclusions are found in Section 8.

## 2. Literature Review

### 2.1. Information System Success Model (D&M Model)

The theory of information system success, which was initially proposed by DeLone and McLean (1992), states that information quality and system quality jointly affect users’ satisfaction with and intention to continue using the system, and satisfaction significantly affects consumers’ willingness to use the system [12]. After 2003, and based on the previous increases in the quality of service, the variables that enhance the universality and practicability of the model used, including the information quality, system quality, and service quality, affect the degree of satisfaction, which affects use intention [13]. The information system success model is also known as the D&M model, in which the information quality comprises a clear statement expression, complete system content, security, relevance, and innovation, and it is easy to understand. To measure the system’s quality, there are indicators, such as usefulness, reliability, responsiveness, etc., to judge whether a system is successfully designed. The indicators that are used to measure service quality include response timeliness, assurance, empathy, etc. High-quality service can contribute to customer retention and improve performance. The D&M model is shown in Figure 1.

The D&M model has been extensively used in research on information system users’ usage intentions, satisfaction, and continuous usage behaviors. Mohammadi pointed out that system and information quality are the main factors driving users’ willingness and satisfaction with e-learning [14]. Tilahun and Fritz showed that system, information, and service quality significantly affected users’ satisfaction and continuous usage intentions [15]. Based on the success theory of information systems, Hsu studied the factors that affect consumers’ willingness to use online group buying [16]. Wang studied the influencing factors of buyback through the D&M model. Through quantitative and qualitative analyses of empirical data, it was concluded that perceived value and satisfaction significantly positively impact buyback intention [17]. Users will decide whether to make the next purchase because of the perceived value and satisfaction improvement.

### 2.2. Technology Acceptance Model (TAM)

The technology acceptance model was proposed by Davis (1989) when he used the rational behavior theory to study users’ acceptance of information systems [18]. It was initially used to conduct extensive research on users’ acceptance behavior regarding information systems. The technology acceptance model proposes two main determinants: perceived usefulness and ease of use. Figure 2 shows the technical acceptance model.

The perceived usefulness refers to a user’s belief that using a specific technology can enhance their work efficiency. When users consider a technology to be practical, they are more likely to accept and use it. Another factor is the perceived ease of use, which refers to the degree of ease and understanding with which a user operates and understands a technology. Technologies with high ease of use are usually more easily accepted and used by users. The TAM primarily focuses on users’ attitudes and behavioral intentions, aiming to explain why individuals accept or reject the use of a particular technology. When studying behavioral intentions, the TAM can be expanded or improved by adding influencing factors [14,19].

The TAM only contains two elements, perceived usefulness and perceived ease of use, and the model does not involve the other key variables that affect consumers’ use intentions. Numerous researchers have expanded upon the model when using it. The most commonly used external variables for expansion include convenience [19,20], expectation confirmation [21], perceived trust [22,23,24,25], subjective norm [25,26,27,28], and enjoyment [26,27]. By incorporating these factors into the study of behavioral intentions, the explanatory power of this article is enhanced.

### 2.3. Integrated Theories and the Proposed Model

The TAM [29,30] and the D&M model [31,32] are effective in predicting and explaining users’ continuous usage behaviors. We take into account the unique characteristics of community group buying platforms and integrate the introduced models to ensure that they align with our goals. For the D&M model, due to the essentially similar information on community group buying platforms, we primarily focus on the service quality and system quality. For the TAM, we will expand upon it. In order to ensure the comprehensiveness and rigor of the study, we introduce three variables (subjective norms, perceived trust, and expectation confirmation) to expand the TAM.

The TAM is an extension of the rational behavior theory, and subjective norms originate from this theory. Therefore, it is reasonable to introduce subjective norms as an external variable in the TAM [26]. Subjective norms determine the behavioral intention [33,34]. The expanded TAM is widely applied, and in the research on behavior intentions in areas such as education [28] and online shopping [9,35], subjective norms have demonstrated their influence. Perceived trust is also an essential factor that can extend the TAM [36]. For example, a dimension of perceived trust can be added to the TAM to better explain users’ acceptance of new technology. In this way, the TAM can provide a more comprehensive explanation of user behaviors and better predict the adoption and diffusion of new technology. In recent years, based on the technology acceptance model, more and more scholars have incorporated trust into it and expanded the TAM, especially in e-commerce [16,23,24]. Expectation confirmation refers to whether the effects after using the platform meet the expectations before use, which is closely related to the continuous use intention. Many studies have introduced this variable for expansion. Bhattacherjee constructed an innovative model, combining expectation validation with perceived usefulness [37]. Brown used polynomial modeling and a response surface analysis to study expectation validation in information systems using technology acceptance models [20]. Malik and Singh integrated the expectation confirmation theory into the technology acceptance model [38]. This combination realizes the integration of expectation validation and the technology acceptance model and obtains a broader range of continuous use model information systems.

Based on the D&M model and the expanded TAM, this paper carries out model innovation. This approach combines both the consumers’ and platform’s perspectives, making the article’s research more comprehensive. The innovative model of this article is shown in Figure 3. The application fields include social platforms, online learning, e-commerce, e-government, mobile search, etc.

## 3. Research Hypothesis Development

### 3.1. Expectation Confirmation

This paper refers to the concept of expectation confirmation defined by Bhattacharjee. It defines expectation confirmation in the context of community group buying as the degree of consumers’ subjective perception after using the community group buying platform and the degree of expectation confirmation before using the community group buying platform. Based on ECM-IT, Bhattacharjee studied the relationships between desired confirmation–perceived usefulness and desired confirmation–satisfaction [37]. The results showed that the two groups were positively correlated.

Joo and Choi studied the factors influencing the willingness to use university institutions’ resources based on the ECM-IT model. In university libraries, the factors affecting students’ intentions to continue using online library resources can be summarized as the positive influence of expectation confirmation on perceived usefulness and user satisfaction [39]. Lin took more than 300 college students as research objects, collected relevant data on the English learning system through a questionnaire survey, and verified the positive correlation between expectation confirmation, user satisfaction, and perceived usefulness [40]. According to the applicable characteristics of expectation confirmation, in the context of a community group buying platform, when consumers feel that the service experience levels of convenience and privacy security enhanced by using the community group buying platform reach or exceed their expectations, they will consider the community group buying platform to be useful and have higher satisfaction with it. To sum up, the following hypothesis, comprising two subhypotheses, is proposed:

**H1a:** 
*Expectation confirmation has a significant positive impact on user satisfaction.*


**H1b:** 
*Expectation confirmation has a significant positive impact on consumers’ continuous usage intentions.*


### 3.2. Perceived Usefulness

Davis first used the concept of perceived usefulness, which provided a new key variable for subsequent research on consumer behavior [18]. Based on this, Bhattacheijee combined the technology acceptance model (TAM) with the expectation confirmation model (ECM) to innovate and build the ECM-IT model, further taking perceived usefulness as the primary variable to measure the continuous usage intention [37].

Akdim showed that perceived usefulness was an important variable affecting satisfaction and continuous usage intention [41]. Some studies discussed behavior regarding the intention to continue using online group buying. The research showed that perceived usefulness significantly impacted user satisfaction and the intention to continue using [35,42]. To sum up, the following hypothesis, comprising two subhypotheses, is proposed:

**H2a:** 
*Perceived usefulness has a significant positive impact on user satisfaction.*


**H2b:** 
*Perceived usefulness has a significant positive impact on consumers’ continuous usage intentions.*


### 3.3. Perceived Ease of Use

Yoon and Kim divided the perceived ease of use according to consumers’ attitudes [21]. When consumers use the community group buying platform, the perceived ease of use refers to the ease of use of time, energy, economy, and other aspects from the beginning of using the platform to the end of the transaction. In other fields, scholars systematically studied the perceived ease of use, pointing out that the perceived ease of use can positively impact the willingness to use and the intention to continue using. Chang explored the factors influencing the intention to continue using a PDA high school English learning system and introduced perceived ease of use into the analysis of the model. It was found that the perceived ease of use has a positive effect on continuous usage intention [19]. Numerous scholars have studied the relationship between perceived usefulness and behavioral intention. They confirmed that the perceived ease of use not only affects satisfaction but also influences purchasing intention [43,44,45]. In summary, the following hypothesis, comprising two subhypotheses, is put forward:

**H3a:** 
*Perceived ease of use has a significant positive impact on user satisfaction.*


**H3b:** 
*Perceived ease of use has a significant positive impact on consumers’ continuous usage intentions.*


### 3.4. Subjective Norms

Subjective norms are often considered significant predictors of an individual’s behavioral intentions. When individuals believe that their social group supports a particular behavior, they are more likely to adopt it [46]. After using the community group buying platform, the surrounding family members and colleagues will inform the subject of their subjective perception, and the subject will form their own will and tendency according to the opinions of others [35]. Due to the social nature of the community group buying platform, platform users are susceptible to subjective norms. These users belong to the same environment, such as the same community or company, and develop behavioral dependencies based on daily trust. Whether online or offline, they will always receive other people’s opinions, further influencing their usage intentions. In summary, the following hypothesis, comprising two subhypotheses, is put forward:

**H4a:** 
*Subjective norms have a significant positive impact on user satisfaction.*


**H4b:** 
*Subjective norms have a significant positive impact on consumers’ continuous usage intentions.*


### 3.5. Perceived Trust

Trust refers to the degree of trust that an individual can generate after making a transaction with other members of society. From the characteristics of community group buying platforms, buyers value the attributes of delivery service platforms and leaders. Competence, friendliness, and integrity thoroughly explained most of the trust effect among more than a dozen characteristics, with no duplications [45]. Kamboj applied the stimulus–organism–response paradigm to explore the factors influencing the intention to use a brand on social media, and the research results showed that perceived trust positively impacted the choice to use a brand [47]. Customer satisfaction with the platform is mainly predicted by trust [8]. This means that after using the community group buying platform, users would receive products and experience the authenticity and reliability of the service, become satisfied, and thus have a sense of trust in the platform, so they would continue to use the community group buying platform, thus forming relationship commitment. According to the survey, as the development of community group buying platforms tends to be centralized, consumers trust community group buying enterprises that survive after adapting to market rules. Therefore, the research on perceived trust in this paper chooses leaders’ friendliness, competence, and integrity. In summary, the following hypothesis, comprising two subhypotheses, is put forward:

**H5a:** 
*Perceived trust has a significant positive effect on user satisfaction.*


**H5b:** 
*Perceived trust has a significant positive effect on consumers’ intentions to continue using.*


### 3.6. System Quality

Stable system support is the most fundamental guarantee of a community group buying platform’s service quality. Jeon analyzed the continuous usage intention of mobile payment applications based on the success model of information systems. The study showed that system quality played a crucial role in this process [31]. System quality directly affects perceived personal benefits and user satisfaction, ultimately determining users’ continuous willingness to consume and provide information. In the D&M model, DeLone and McLean found through an investigation that system quality significantly positively impacts users’ willingness to use the system and their satisfaction with it [13]. In a study on users’ willingness to exchange information in virtual communities, Zheng found and proved that personal interest and user satisfaction are directly affected by system quality and information, and thus, they affect or even determine users’ final consumption behaviors and willingness to continue providing information [48]. In a community group buying platform, system quality, satisfaction, and continuous usage intention also have similar relationships with the above content. When users experience conditions such as a fast system response speed, stable operation, and no lag (i.e., better system quality), they will be more satisfied with the platform and, thus, more willing to continue using it. To sum up, this paper puts forward the following hypothesis, comprising two subhypotheses:

**H6a:** 
*System quality has a significant positive impact on user satisfaction.*


**H6b:** 
*System quality has a significant positive impact on consumers’ continuous usage intentions.*


### 3.7. Service Quality

Service quality refers to the auxiliary services provided by e-commerce platforms and merchants, such as the shopping environment, product quality, and other supporting services, for consumers’ shopping experiences [49]. This definition has been supported and cited by many scholars in the academic world. Based on the above description, this paper proposes the following definition of service quality in the community group buying environment: service quality is the difference between the user’s expectation of service and the user’s actual service experience before experiencing the group buying service on the community platform. In the D&M model proposed by DeLone and McLean, a positive correlation between service quality and customer satisfaction was proposed. Later, Padlee also proposed a model of the relationship between service quality, customer satisfaction, and behavioral intention and demonstrated a positive correlation between the three [50]. Aliman used convenient sampling technology and a multiple regression model to test the hypothesis in their research on mobile group buying users, and the results showed that service quality was positively correlated with expected behavior, and service factors were positively correlated with user satisfaction [51]. In the context of community group buying services, if the goods purchased by consumers are of good quality, the delivery is timely, the after-sales service platform responds quickly, and the needs of the consumers are handled well, consumers will perceive the platform to have a high service quality, thus improving their satisfaction with the platform and enhancing their willingness to use it. In summary, the following hypothesis, comprising two subhypotheses, is put forward:

**H7a:** 
*Service quality has a significant positive effect on user satisfaction.*


**H7b:** 
*Service quality has a significant positive effect on consumers’ continuous usage intentions.*


### 3.8. User Satisfaction

One of the most essential measures of user behavior is user satisfaction. According to Bhattacharjee’s continuous use model of information systems, satisfaction refers to how satisfied users are when they experience or use the system. Wang proved in the D&M model that the improvement of satisfaction directly impacts the growth of the intention to continue using [17]. Bhattacherjee found and confirmed a positive correlation between satisfaction and users’ intentions to continue using information systems in his study on the literature relational system. Delone and McLean also confirmed the same conclusion through their study on the D&M model that user satisfaction has a positive impact on users’ continuous usage intentions. After users experience functions, processes, shopping, and other functions, their satisfaction with the community group buying platform directly affects their intentions to continue using it. As users’ satisfaction with a community group buying platform increases, users’ willingness to use the community group buying platform will also increase. To sum up, the following hypothesis is proposed:

**H8:** 
*User satisfaction has a significant positive impact on consumers’ continuous usage intentions.*


### 3.9. Continuous Usage Intention

Continuous usage intention refers to a consumer’s attitude and tendency toward whether they will continue to use a particular product or service in the future. Continuous usage intention is a kind of subjective attitude and consumer behavior intention. After using the platform for the first time, consumers intend to continue using the platform. We will continue to use this platform over other platforms in the future and will increase the frequency of use of this platform [52]. Combined with the service characteristics of a community group buying platform, this paper summarizes and puts forward a new definition of continuous use intention. Based on the experience and use of the community group buying platform, users are willing to continue to use the current community group buying platform, have a preference to use the existing community group buying platform, and are willing to explore new functions in the community group buying platform under the same conditions.

### 3.10. Mediating Effect

Keeney pointed out that if the product quality meets the customers’ expectations, the customers will often consider online shopping centers beneficial and continue to visit them [53]. In the service context of a community group buying platform, when consumers feel that the service experience levels of convenience, privacy, and security improved by using a community group buying platform reach or are higher than expected, they will consider the platform to be helpful and have higher satisfaction levels. In addition, Yeong studied consumers’ purchasing behaviors in catering services and found that subjective norms significantly positively affected perceived usefulness [44]. In summary, the following hypotheses are put forward:

**H9:** 
*Expectation confirmation has a significant positive impact on consumers’ perceived usefulness.*


**H10:** 
*Subjective norms have a significant positive impact on consumers’ perceived usefulness.*


## 4. Research Method

### 4.1. Design of Research Scheme

According to the diversity of the nine variables in this paper, the survey method is more suitable with a multi-item scale than a single scale. In addition, the Likert scoring method has been widely used in academic circles, which is enough to prove the applicability of the five-point scoring method. The selection of project content and variable factors is based on previous relevant academic research. In the process of making the scale, in addition to considering professionalism, the differences between the community group buying platform and other studies were weighed, and the content details were improved based on the critical points used by the consumers. The details are shown in Table 1.

### 4.2. Data Collection and Sample Profile

In China, cities in Liaoning Province and Hubei Province were selected for research. The geographical distribution of cities in the two provinces is large, which can better represent the different characteristics of community group buying users. In order to collect data that reflect the community’s aggregation, the survey channels are distributed through various regional property owner groups and merchant groups. There are 418 valid questionnaires, which are consistent with the research objective of this paper. Among the 559 questionnaires collected, 418 were selected that fit the research scope, accounting for 74.78%. Regarding gender, women account for 86.4%, men account for 13.6%, and women almost occupy the whole share. Regarding marital status, married individuals account for 91.5%, unmarried individuals account for 8.5%, and married individuals almost accounted for the whole share. Regarding the age distribution, the proportion between 30 and 39 years old is the highest, accounting for 38.5%, followed by 40 to 49 years old, accounting for 27.7%, indicating that the age distribution is concentrated between 30 and 49 years old. In terms of housing types, fourth-tier cities and those below occupy the highest proportion, accounting for 56.7%, followed by third-tier cities, accounting for 20.1%. It can be seen that people suitable for community group buying are mainly concentrated in sinking cities. In terms of income level, CNY 5000–8000 and CNY 8000–10,000 accounted for the highest proportions, 33.1% and 36.1%, respectively. It can be seen that consumers who use community group buying generally have a specific economic basis. In general, most community group purchase users are married middle-aged women in the lower cities with one particular economic basis, which means that the survey results are consistent with reality.

## 5. Statistical Analysis

### 5.1. Reliability Test

This study examines the reliability of the collected data through SPSS 24.0, and the results are shown in Table 2.

The overall Cronbach’s alpha coefficient value of the scale and the nine dimensions are all greater than 0.7. This indicates that the content of the designed survey questionnaire has high consistency, meaning that the reliability is excellent. The correlation between the items in the questionnaire and the overall score is higher than 0.3. Most of the items are significantly related to the overall score.

### 5.2. Validity Test

A validity analysis, which is a crucial step in the research methodology, is primarily used to assess the degree to which a measuring tool or method can accurately measure the intended subject matter. In this section, we conducted a preliminary analysis of the validity of the data using SPSS 24.0, with the analysis results presented in Table 3.

The KMO value is used to judge whether the validity is valid, the covariance value is used to exclude unreasonable research projects, the variance interpretation rate value represents the level of information extraction, and the factor load coefficient measurement factor (dimension) corresponds to the project. As can be seen from Table 3, the standard value of all research items is greater than 0.4, the KMO value is 0.884, which is more significant than 0.7, and the interpretation rate of cumulative variance after rotation is 72.626% > 60%, indicating that information about the research items can be effectively extracted.

According to the meaning of the problem in the scale and the rotating component matrix, the load value is greater than 0.5, indicating that it can be analyzed as an essential item. The results show that the load value of each item in each dimension is greater than 0.5, the results obtained by rotating the component matrix are consistent with the scale and dimension, and the questionnaire has high validity and effectiveness.

In this paper, the Harman single-factor test was used to test the homogeneity of the data. The specific methods are as follows: a factor analysis is carried out on all scale items together, the factor load matrix is checked without rotation, and the size of homologous deviation is determined according to the first principal component precipitated in the matrix. As the first principal component is 30.81% < 40%, the homology deviation is not serious.

### 5.3. Structural Equation Model Testing

Structural equation models (SEMs) can reflect the direct and indirect relations between latent variables under the premise of error variables. Therefore, this study uses SEM to verify the overall model constructed with nine latent variables and the path coefficients are tested.

As can be seen from Table 4, in the model of this study, CMIN/DF, NFI, IFI, TLI, CFI, GFI, RMSEA, CFI, and the other model fitness indicators all meet the standards, and the model fitness is good. The path test results are shown in Table 5.

The results indicate that expectation confirmation significantly positively affects perceived usefulness, user satisfaction, and continuous usage intention, and the path coefficients are 0.360, 0.191, and 0.181. Perceived usefulness has a significant positive effect on user satisfaction and continuous usage intention, with path coefficients of 0.258 and 0.157. Subjective norms have a significant favorable influence on perceived usefulness, and the path coefficient is 0.372. At the same time, subjective norms also have a significant positive effect on continuous usage intention, and the path coefficient is 0.156. Perceived trust has a significant positive impact on user satisfaction and continuous usage intention, and the path coefficients are 0.231 and 0.144. Perceived ease of use has a significant positive effect on user satisfaction and continuous usage intention, and the path coefficients are 0.167 and 0.133. Service quality has a positive effect on user satisfaction and continuous usage intention, with path coefficients of 0.141 and 0.119. User satisfaction is significantly positively correlated with continuous usage intention, with a path coefficient of 0.133.

In this paper, the number of random repeated samples was set to 2000, the confidence was 95%, and the bias-corrected bootstrap estimation method was used to test and analyze. According to the standard of the confidence interval method, if the path does not contain 0 in the bias-corrected confidence interval, on this basis, if the confidence interval of the direct effect contains 0, this indicates that the direct effect does not exist, and it is a complete intermediary effect. If the confidence interval of the direct effect does not contain 0, this indicates that the direct effect exists, and it is a partial intermediary effect. The test results of the intermediate variables of each path obtained by AMOS24.0 are shown in Table 6.

Through an analysis of each intermediary path of the model, the confidence interval and test significance of the indirect effect deviation correction corresponding to each path are shown in the table above.

As can be seen from Table 6, under the path of “Service quality → User satisfaction → Continuous usage intention”, the deviation correction CI of the mediating effect based on user satisfaction is [0.001,0.056], the interval does not contain 0, and *p* < 0.05. This indicates that the indirect effects of this path are significant, and there exists an intermediary effect. Using the same method, we found that the indirect effect under the path of “system quality → user satisfaction → continuous use intention” is not significant, and there is no intermediary effect. In addition, there is no mediating effect in the path of “subjective norm → user satisfaction → continuous usage intention, subjective norm → perceived usefulness → continuous usage intention”.

In this model, expectation confirmation indirectly affects the intention to continue using through perceived usefulness and user satisfaction. Subjective norms indirectly affect the intention to continue using through perceived usefulness and user satisfaction. Perceived usefulness, subjective norms, perceived trust, service quality, system quality, and perceived ease of use indirectly affect the continuous usage intention through the mediating variable of user satisfaction.

As seen from Table 7, the above paths indicate that service quality, perceived trust, perceived ease of use, perceived usefulness, and expectation confirmation can not only directly predict user satisfaction but also predict continuous usage intention through the mediating effect of user satisfaction.

In view of the related research hypothesis proposed above, the input test of the hypothesis is carried out through the variance analysis of SPSS24.0 and the structural equation analysis of AMOS24.0. The results are shown in Table 8.

The results show that hypotheses H4a, H6a, and H6b are invalid. The model of consumers’ intentions to continue using the community group buying platform after verification is shown in Figure 4.

## 6. Discussion and Implications

### 6.1. Discussion

This study’s findings suggest that expectation confirmation has a significant positive impact on perceived usefulness, user satisfaction, and continuous usage intention. This is consistent with the research findings of Joo [39] and Lin [40]. Therefore, in the long-term development of community group purchase platforms, in order to improve user retention and solve the problem of repeat purchases, it is necessary to pay special attention to consumer expectations and consumption habits. Subjective norms also have a significant positive impact on continuous usage intention. This result has also been confirmed in other studies [54,55,56]. However, the direct impact of subjective norms on satisfaction is not tested in the model. This is different from previous academic research findings, which can be attributed to the rapid development of community group purchasing and the changing competitive landscape. Perceived trust has a significant positive impact on satisfaction and continuous usage intention. This is consistent with the research conducted by Ruiz-Herrera [36] and Sasonko [57] but different from the study by Tian and Chan [46]. The other results are due to the different types of platforms studied, with differences in the use of electronic wallets and community group purchasing platforms. Perceived trust is crucial for the intention to continue using. The perceived ease of use has a significant positive impact on satisfaction and continuous usage intention. The research conducted by Mohammadi [14] and Goundar [22] supports this conclusion. However, some studies have shown that the perceived ease of use does not have a significant positive impact on behavioral intention or that it has a very limited impact on willingness [55,58]. Our explanation for this is that the shopping method of community group purchasing platforms is different from other e-commerce or supermarkets. The online service and appointment-based self-pickup shopping method alleviate consumers’ time constraints. With affordable goods and personalized services available at the self-pickup point at their doorstep, consumers’ perceived convenience will be internalized as user satisfaction, thus influencing their future purchases.

Service quality and system quality are variables introduced in this study within the D&M model. The research results of this article indicate that service quality positively affects user satisfaction and continuous usage intention. This is consistent with the existing research conclusions on this topic [14,46,49,51]. However, system quality has not passed all the tests for its influence on continuous usage intention. This is in contrast to the existing conclusions [10,14,49]. Therefore, the relevant community group purchasing platform operation departments should pay attention to the impact of service quality on consumer satisfaction and improve the service quality of various aspects of community group purchasing, for example, by improving the environment of self-pickup points and providing timely and efficient after-sales service for consumers. In the hypothesis testing results, system quality did not have a direct and indirect significant positive effect on the continuous use intention of community group purchasing. This may be because consumers’ concerns focus on service quality and whether it can better meet their needs. The problems of system quality such as slow systems, stuttering, and unclear video and audio are very rare in system quality, and the system control of community group purchasing platforms still shows the stability of the current internet level. In addition, as a system problem, as long as it does not affect the effect of ordering and after-sales service, consumers may still choose to continue using the services of community group purchasing platforms with the continuous upgrading of the system.

User satisfaction is a key variable in the research model proposed in this article. The study has shown that there is a significant positive correlation between user satisfaction and continuous usage intention. At the same time, the results of the mediation test show that satisfaction plays a mediating role in the relationship between expectation confirmation, perceived usefulness, service quality, perceived ease of use, and perceived trust in the continuous use intention of community group purchasing platforms. Most studies have also confirmed this conclusion [17,48,59], and satisfaction has a significant effect on continuous use intention. In the field of community group purchasing, after experiencing functions such as shopping, a user’s satisfaction with the community group purchasing platform directly affects the user’s continuous use intention. As the user’s satisfaction with a community group purchasing platform increases, their willingness to use it also increases. This suggests that the planning and operation departments of community group purchasing platforms need to deeply understand user satisfaction and propose corresponding countermeasures to improve user satisfaction. This is significantly effective in maintaining the user’s intention to continue using the community group purchasing platform.

### 6.2. Theoretical Implications

The literature review shows that although community group purchasing platforms have received significant attention from scholars, the current research is primarily descriptive, focusing on content creation and platform revenue models, and it lacks discussions on users’ willingness to use the platforms, especially from the consumers’ perspectives on the continuous use of the platforms. Based on the existing literature and the characteristics of community group purchasing platforms, this article proposes an innovative and widely applicable theoretical model, which uses the extended TAM and the D&M model to explore the factors affecting users’ continuous use of community group purchasing platforms from multiple perspectives. This can enrich the research focusing on users’ subjective experiences while using community group purchasing platforms, reduce the influence of the platform’s perspective, and further excavate the influencing factors of users’ willingness to continue using at the individual level.

The extension of the TAM and its integration with other models are indeed a trend [60]. The establishment of this model combines the extended TAM and the DM model, which includes factors from both the consumer and platform perspectives. On the one hand, this provides new ideas and perspectives for the extension and integration of theoretical models. On the other hand, it enhances the explanatory power and comprehensiveness of the article.

### 6.3. Practical Implications

This paper summarizes the main factors that affect consumers’ continuous use of community group buying platforms. Based on the results of the empirical analysis, the platform can better identify and predict the psychological and behavioral patterns of users, take more targeted measures to retain users, improve user loyalty and satisfaction, develop more attractive content for users, encourage users to use the community group buying platform more frequently, and even generate more traffic. In addition, it also has a specific reference value for other social e-commerce service platforms similar to community group buying platforms. This paper proposes the following management implications:

First, this study shows that among the direct influencing factors of consumers’ intentions to continue using, expectation confirmation, perceived usefulness, and perceived ease of use significantly affect consumers’ intentions to continue using. The platform can reduce the after-sales service audit time and enhance consumers’ perceptions of ease of use, promote the qualification of commodity channels, ensure the continuity and security of supply, ensure the transparency and openness of online shopping, provide preferential prices for group buying, and increase consumer awareness of the usefulness of platforms [61].

Second, service organizations should focus on service quality and the environment to convince consumers to continue using [62]. Platform operators need to continuously optimize the service quality of the platform, improve the transaction environment and offline environment, and improve user satisfaction [63]. Platform operators need to (1) ensure the quality of goods, improve distribution efficiency, and meet consumer demands; (2) improve the environment for consumers to pick up goods and distinguish between the daily necessities and fresh areas; (3) use big data to provide diversified and accurate services for consumer needs; and (4) smooth user complaint channels and efficiently solve pre-sales and after-sales disputes.

Third, trust can provide stable psychological expectations for community group buying platforms and consumers, thereby reducing transaction costs caused by information asymmetry. The optimization of trust for community group buying platforms that focus on service is actually the optimization of people. Staff involved in procurement, sorting, transportation, sales, and other links, as well as platform customer service, should all focus on improving professionalism, improving service attitudes, and establishing professional ethics [64]. The platform can choose high-quality brands as its characteristic brands from the channel end, establish a brand image, improve brand recognition, cultivate consumers’ choice loyalty, and improve their willingness to continue to use the platform.

Fourth, for consumers, information is their privacy and identity [45]. In order to meet the needs of refined and featured services, users are willing to synchronize their information to the platform, but this does not mean that the platform can do whatever it wants. The community group buying platform should strengthen the management of information security when it grasps user information. Specifically, managers can choose to clarify the scope of privacy authorization, regularly access information security issues, and guarantee the independence of the system. This will encourage users to use the security platform actively. Increasing the frequency of use will improve the platform’s information data analysis efficiency, provide more personalized services, strengthen consumers’ consumption habits, and form a closed loop.

Fifth, the convenience of the platform to provide services is essential. Community group buying platforms should not only be limited to commodity innovation but should also conform to the technological progress of the times. The application of cloud computing, the Internet of Things, and even VR technology in life scenes allows consumers to experience convenience to the maximum extent, but also allows them to have fun and participate. In the future, online selection and VR and AR technology can be used to feel the source of goods [65], live broadcasts can be used to popularize the system operation of the supply chain, and the linkage of intelligent terminal equipment can be carried out to achieve unmanned automated distribution.

## 7. Limitations and Future Research Directions

Due to the limitations of time and region, this study has some limitations and deficiencies. The first limitations are the study’s sample size and regional scope. Second, this study focuses on subjective perception to reflect consumers’ behavioral intentions, so relevant factors in the model are all subjective factors of the consumers. Third, the research object needs to be refined. Consumers are not an identical whole, and consumer groups in different regions have different characteristics. For the above problems, the collection scope will be expanded in the follow-up study on similar problems to achieve comprehensive and accurate results. In addition, future relevant studies can enrich the selection range of factors, such as price, platform authentication, and other factors, together with the subjective factors in this paper as objective perspectives.

## 8. Conclusions

The research aim of this article is to verify the influences of different factors on users’ continuous use of community group buying platforms, thus exploring how platforms can maintain their competitive advantages and sustainable development under the challenges of intense competition and user churn. According to the changing trend of the community group buying platform market and the change in consumers’ usage habits, this paper constructs a new model system through the innovation of the D&M model and the expanded TAM. The data are collected through questionnaire surveys to test relevant hypotheses. The specific conclusions are as follows: This study integrates two models to validate the degree and mechanism of influence of factors such as expectation confirmation, perceived usefulness, perceived ease of use, subjective norm, perceived trust, service quality, system quality, and user satisfaction on the intention to continue using community group purchase platforms. This study found that perceived usefulness, perceived ease of use, expectation confirmation, perceived trust, and service quality are crucial factors in enhancing user satisfaction and promoting continuous usage intentions. Subjective norms directly and significantly affect consumers’ intentions to continue using the platform. Perceived usefulness, perceived ease of use, service quality, perceived trust, and expectation confirmation significantly and indirectly affect continuous usage intention through user satisfaction. The effect of system quality on the continuous usage intention is not significant. The conclusion of this article provides a basis for offering management insights.

## Figures and Tables

**Figure 1 behavsci-13-00941-f001:**
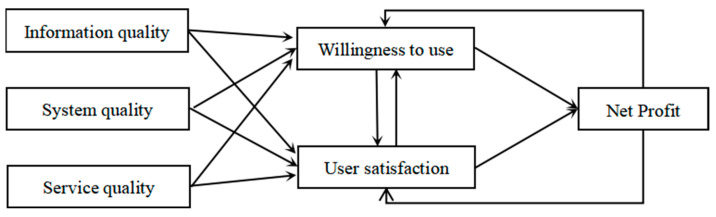
D&M model.

**Figure 2 behavsci-13-00941-f002:**
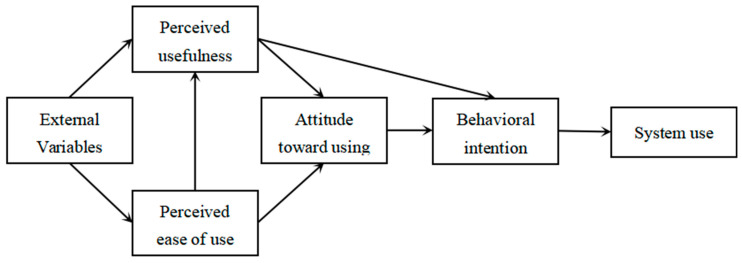
Technology acceptance model.

**Figure 3 behavsci-13-00941-f003:**
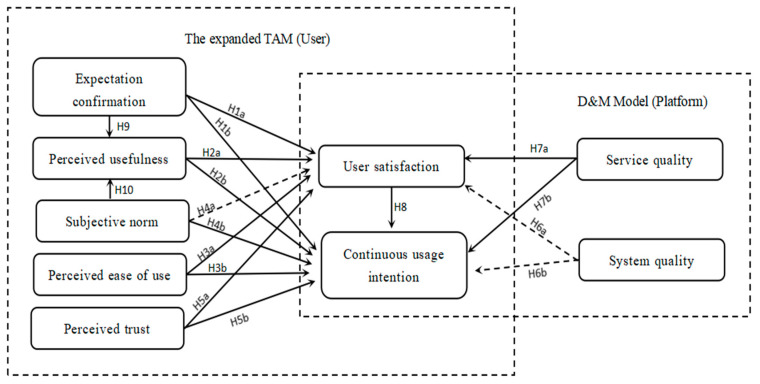
Theoretical model.

**Figure 4 behavsci-13-00941-f004:**
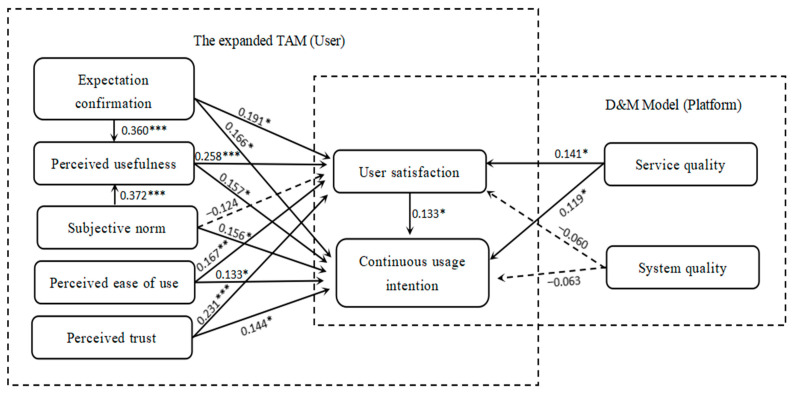
Revised model. Note: The solid line is significant; the dotted line is not significant. (* *p* < 0.05; ** *p* < 0.01; *** *p* < 0.001).

**Table 1 behavsci-13-00941-t001:** Questionnaire scale variable measurement content and reference sources.

Variable Factor	Item Content	Reference Source
Perceived usefulness	1. The community group buying platform provides a wide range of commodities that can meet my daily needs. 2. I can buy cost-effective, cheap, and cost-effective goods on the community group buying platform. 3. The goods delivery service of community group purchase should satisfy me. 4. Using the community group buying platform can increase my choice space and improve my quality of life.	Bhattacharjee [37] Davis [18] Lee [42]
Perceived trust	1. My brand trust in the community group buying platform makes me more interested in using it for shopping. 2. I think the leader is very concerned about my interests and needs 3. The leader tries to be fair in his dealings with others. 4. I think the leader is capable of doing his job well.	Shiau [8] Kamboj [47] Alzaidi [45]
Perceived ease of use	1. I know how to complete shopping, pick-up, return, and exchange with a community group buying platform on a mobile phone, which is easy to master. 2. I think using a community buying platform can save a lot of time and effort. 3. I think the after-sale service of the community group buying platform solves my concerns to a large extent.	Yoon [21] Chang [19] Lim [43]
Subjective norms	1. My friends and family are happy to use the community group buying platform. 2. I think a lot of people are using community group buying platforms. 3. People who have influenced me greatly support my use of the community buying platform.	Zhai and Zhang [35] Tian and Suki [46]
System quality	1. I think the system has a strong response processing capacity. 2. I think the community group buying platform has high privacy and security.	Jeon [31] Zheng [48]
Service quality	1. My application for services on the community group buying platform was quickly responded to. 2. The leader can help the platform solve problems promptly.	Han [49] Padlee [50] Aliman [51]
Expectation confirmation	1. The “leader & self-pickup” service provided by the community group buying platform is reasonable. 2. The goods I bought met my expectations after using the community group buying platform. 3. After using the community group buying platform, I felt that the levels of platform services, content, and other aspects were higher than expected.	Bhattacharjee [37] Joo [39] Lin [40]
User satisfaction	1. I am satisfied with using the community group buying platform for shopping. 2. I am happy with the functionality of using the community group buying platform. 3. The life brought by the community group buying platform helps me feel satisfied.	Wang [17] Bhattacharjee [37] Delone& McLean [13]
Continuous usage intention	1. I am happy to recommend community group buying platforms to my friends. 2. I would like to continue using the community group buying platform. 3. Under the same conditions, I would like to prioritize the community group buying platform.	Bhattacharjee [52]

**Table 2 behavsci-13-00941-t002:** Reliability analysis. (The numbers following the variables correspond to the measurement items in Table 1).

Cronbach Reliability Analysis				
Variables	Total Correlation of Correction Items	The α Coefficient of the Term Has Been Deleted	Cronbach α Coefficient	Total Cronbach α Coefficient
Perceived usefulness 1	0.563	0.754	0.789	0.908
Perceived usefulness 2	0.680	0.693		
Perceived usefulness 3	0.577	0.747		
Perceived usefulness 4	0.573	0.749		
Perceived trust 1	0.530	0.724	0.763	
Perceived trust 2	0.579	0.699		
Perceived trust 3	0.589	0.693		
Perceived trust 4	0.555	0.711		
Perceived ease of use 1	0.619	0.743	0.796	
Perceived ease of use 2	0.622	0.742		
Perceived ease of use 3	0.689	0.671		
Subjective norm 1	0.671	0.651	0.781	
Subjective norm 2	0.569	0.756		
Subjective norm 3	0.634	0.690		
System quality 1	0.630	-	0.765	
System quality 2	0.630	-		
Service quality 1	0.665	-	0.783	
Service quality 2	0.665	-		
Expectation confirmation 1	0.669	0.648	0.781	
Expectation confirmation 2	0.635	0.687		
Expectation confirmation 3	0.561	0.763		
User satisfaction 1	0.785	0.855	0.884	
User satisfaction 2	0.761	0.850		
User satisfaction 3	0.827	0.807		
Continuous usage intention 1	0.735	0.851	0.877	
Continuous usage intention 2	0.727	0.858		
Continuous usage intention 3	0.828	0.766		

**Table 3 behavsci-13-00941-t003:** Efficiency analysis.

Results of Validity Analysis										
Variables	Factor Load Coefficient									Common Degree (Variance of Common Factor)
	Factor 1	Factor 2	Factor 3	Factor 4	Factor 5	Factor 6	Factor 7	Factor 8	Factor 9	
Perceived usefulness 1	0.599									0.542
Perceived usefulness 2	0.841									0.771
Perceived usefulness 3	0.671									0.610
Perceived usefulness 4	0.688									0.617
Perceived trust 1			0.627							0.538
Perceived trust 2			0.735							0.639
Perceived trust 3			0.810							0.708
Perceived trust 4			0.667							0.596
Perceived ease of use 1					0.738					0.687
Perceived ease of use 2					0.746					0.690
Perceived ease of use 3					0.863					0.803
Subjective norm 1							0.822			0.758
Subjective norm 2							0.752			0.648
Subjective norm 3							0.791			0.712
System quality 1									0.845	0.803
System quality 2									0.884	0.821
Service quality 1								0.859		0.823
Service quality 2								0.880		0.829
Expectation confirmation 1						0.841				0.770
Expectation confirmation 2						0.769				0.709
Expectation confirmation 3						0.706				0.617
User satisfaction 1		0.858								0.834
User satisfaction 2		0.788								0.796
User satisfaction 3		0.867								0.864
Continuous usage intention 1				0.838						0.808
Continuous usage intention 2				0.694						0.748
Continuous usage intention 3				0.851						0.868
Cumulative variance interpretation rate % (after rotation)	72.63%									-
KMO	0.884									-
Bartlett	5030.334									-
df	351									-
*p*	0									-

**Table 4 behavsci-13-00941-t004:** Fitting coefficient of structural model.

Model Fit Coefficients								
CMIN	df	CMIN/DF	NFI	IF	TLI	CFI	GFI	RMSEA
468.065	292	1.603	0.909	0.964	0.956	0.963	0.926	0.038
Suggested value		<3	>0.8	>0.9	>0.8	>0.9	>0.8	<0.08

**Table 5 behavsci-13-00941-t005:** Path test results. (*** represents *p* < 0.001).

Pathway Test Results									
Path	Relationship Path between Variables			Non-Standardized Regression Coefficient	Standardized Regression Coefficient β	Standard Error	*t*	*p*	Pathway Test Results
Path 1	Perceived usefulness	←	Expectation confirmation	0.334	0.360	0.064	5.207	***	Support
Path 2	Perceived usefulness	←	Subjective norms	0.312	0.372	0.057	5.445	***	Support
Path 3	User satisfaction	←	Expectation confirmation	0.303	0.191	0.119	2.536	0.011	Support
Path 4	User satisfaction	←	Subjective norms	−0.178	−0.124	0.102	−1.751	0.080	Nonsupport
Path 5	User satisfaction	←	Perceived usefulness	0.441	0.258	0.122	3.624	***	Support
Path 6	User satisfaction	←	Perceived ease of use	0.202	0.167	0.076	2.668	0.008	Support
Path 7	User satisfaction	←	Perceived trust	0.353	0.231	0.106	3.326	***	Support
Path 8	User satisfaction	←	System quality	−0.079	−0.060	0.073	−1.083	0.279	Nonsupport
Path 9	User satisfaction	←	Service quality	0.159	0.141	0.061	2.592	0.010	Support
Path 10	Continuous usage intention	←	Expectation confirmation	0.181	0.166	0.078	2.315	0.021	Support
Path 11	Continuous usage intention	←	Subjective norms	0.154	0.156	0.067	2.307	0.021	Support
Path 12	Continuous usage intention	←	Perceived usefulness	0.185	0.157	0.081	2.289	0.022	Support
Path 13	Continuous usage intention	←	User satisfaction	0.092	0.133	0.041	2.251	0.024	Support
Path 14	Continuous usage intention	←	Perceived ease of use	0.111	0.133	0.050	2.220	0.026	Support
Path 15	Continuous usage intention	←	Perceived trust	0.151	0.144	0.070	2.156	0.031	Support
Path 16	Continuous usage intention	←	System quality	−0.057	−0.063	0.047	−1.209	0.227	Nonsupport
Path 17	Continuous usage intention	←	Service quality	0.093	0.119	0.040	2.298	0.022	Support

**Table 6 behavsci-13-00941-t006:** Test results for path intermediate variables.

Mediation Effect Test				
Path	Mediating Variable	Indirect Effect		
		Boot CI Lower Limit	Boot CI Upper Limit	*p*
Service quality → User satisfaction → Continuous usage intention	User satisfaction	0.001	0.056	0.031
System quality → User satisfaction → Continuous usage intention	User satisfaction	−0.034	0.004	0.174
Perceived trust → User satisfaction → Continuous usage intention	User satisfaction	0.003	0.076	0.029
Perceived ease of use → User satisfaction → Continuous usage intention	User satisfaction	0.002	0.065	0.031
Subjective norms → User satisfaction → Continuous usage intention Subjective norms → Perceived usefulness → Continuous usage intention Subjective norms → Perceived usefulness → User satisfaction → Continuous usage intention	User satisfaction and perceived usefulness	−0.007	0.134	0.084
Perceived usefulness → User satisfaction → Continuous usage intention	User satisfaction	0.004	0.091	0.026
Expectation confirmation → User satisfaction → Continuous usage intention Expectation confirmation → Perceived usefulness → Continuous usage intention Expectation confirmation → Perceived usefulness → User satisfaction → Continuous usage intention	User satisfaction and perceived usefulness	0.033	0.180	0.002
Expectation confirmation → Perceived usefulness → User satisfaction	Perceived usefulness	0.034	0.183	0.001

**Table 7 behavsci-13-00941-t007:** Proportion of mediating effect.

Path	Effect	Effect Value	Relative Effect Value
Service quality → User satisfaction → Continuous usage intention	Total effect	0.138	
	Direct effect	0.119	86.32%
	Mediating effect	0.019	13.77%
Perceived trust → User satisfaction → Continuous usage intention	Total effect	0.174	
	Direct effect	0.143	82.18%
	Mediating effect	0.031	17.82%
Perceived ease of use → User satisfaction → Continuous usage intention	Total effect	0.155	
	Direct effect	0.133	85.81%
	Mediating effect	0.022	14.19%
Perceived usefulness → User satisfaction → Continuous usage intention	Total effect	0.191	
	Direct effect	0.157	82.20%
	Mediating effect	0.034	17.80%
Expectation confirmation → User satisfaction → Continuous usage intention Expectation confirmation → Perceived usefulness → Continuous usage intention Expectation confirmation → Perceived usefulness → User satisfaction → Continuous usage intention	Total effect	0.260	
	Direct effect	0.166	63.85%
	Mediating effect	0.094	36.15%
Expectation confirmation → Perceived usefulness → User satisfaction	Total effect	0.284	
	Direct effect	0.191	67.25%
	Mediating effect	0.093	32.75%

**Table 8 behavsci-13-00941-t008:** Assumed test results.

Number	Hypothetical Content	Inspection Result
**H1a**	*Expectation confirmation has a significant positive impact on user satisfaction.*	Support
**H1b**	*Expectation confirmation has a significant positive impact on consumers’ continuous usage intentions.*	Support
**H2a**	*Perceived usefulness has a significant positive impact on user satisfaction.*	Support
**H2b**	*Perceived usefulness has a significant positive impact on consumers’ continuous usage intentions.*	Support
**H3a**	*Perceived ease of use has a significant positive impact on user satisfaction.*	Support
**H3b**	*Perceived ease of use has a significant positive impact on consumers’ continuous usage intentions.*	Support
**H4a**	*Subjective norms have a significant positive impact on user satisfaction.*	Nonsupport
**H4b**	*Subjective norms have a significant positive impact on consumers’ continuous usage intentions.*	Support
**H5a**	*Perceived trust has a significant positive impact on user satisfaction.*	Support
**H5b**	*Perceived trust has a significant positive impact on consumers’ continuous usage intentions.*	Support
**H6a**	*System quality has a significant positive impact on user satisfaction.*	Nonsupport
**H6b**	*System quality has a significant positive impact on consumers’ continuous usage intentions.*	Nonsupport
**H7a**	*Service quality has a significant positive impact on user satisfaction.*	Support
**H7b**	*Service has a significant positive impact on consumers’ continuous usage intentions.*	Support
**H8**	*User satisfaction has a significant positive impact on consumers’ continuous usage intentions.*	Support
**H9**	*Expectation confirmation has a significant positive impact on consumers’ perceived usefulness.*	Support
**H10**	*Subjective norms have a significant positive impact on consumers’ perceived usefulness.*	Support

## Data Availability

The raw data supporting the conclusions of this article will be made available by the authors, without undue reservation.

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
