# Peer review of "Determinants of Continuous Usage Intention in Community Group Buying Platform in China: Based on the Information System Success Model and the Expanded Technology Acceptance Model"

_behavsci, 2023, doi:10.3390/bs13110941_

Round 1
Reviewer 1 Report
Comments and Suggestions for Authors
The paper is well-organized with clearly defined sections, making it easier to follow the argument. The authors have thoroughly reviewed existing literature, which adds depth to their study. The paper addresses a gap in the literature and provides new insights that have broader implications for the field.
However, this draft has the following drawbacks:
The research objectives could be more explicitly stated to give the reader a better understanding of the research focus. The methodology section lacks detail, making it difficult to replicate the study. The statistical analyses could be more comprehensive. For instance, the paper could benefit from advanced statistical tests. The discussion section could provide a more nuanced interpretation of the findings, including limitations and future research directions. Some citations are either outdated or irrelevant to the current research. Updated and more relevant literature should be included.
This study adopted an overused model: TAM. The authors should provide more insight viewpoints to this adoption. For the statistics analysis, Finally, the author should provide a section for research implications in a formal way.
Reviewer 2 Report
Comments and Suggestions for Authors
The title of the paper should clearly indicate the focus of the research. A revised title is a must, to represent the alignment between the paper and the title. The abstract provides a succinct overview of the paper's objectives and findings.
The introduction effectively introduces the concept of community group buying and its significance in the context of e-commerce. It successfully sets the stage for the research, but it could be enhanced by explicitly stating the research objectives and highlighting the novelty of the study. Additionally, the reference formatting should be reviewed and corrected.
The literature review provides a background on community group buying and its previous research. However, it could be improved by organizing the information more coherently and by discussing the gaps in the existing literature that this study aims to address also add more recent references. The hypotheses are well-formulated and logically derived from the literature review.
Additionally, thorough proofreading and formatting adjustments are necessary to ensure accuracy and consistency in the references.
Comments on the Quality of English LanguageEnough scope for enhancing the quality of English in the paper.
Reviewer 3 Report
Comments and Suggestions for Authors
The authors address interesting topic in the field of Group Buying platform by empirically examined the determinants of continuous usage intention based on a theory based integrated model. Having said that, I would like to give comments and feedback which expectedly improve the quality of the manuscript.
The title states that this research is combining two theories D&M and TAM. However, the literature review section shows that the present study is combining three theories namely: 2.1. Information system success model (D&M model); 2.2. The theory of planned behavior; and 2.3. Technology acceptance model (TAM). First, this should be highlighted in the title. Second, I recommend adding a new section as explain the integrated theories as following: 2.4. Integrated theories and the proposed model.
The hypothesis should be illustrated in Figure 3. Theoretical model. As flowing: H1, H2, H3, H4……
Table 3. Efficiency analysis, is too big and not clear to the reader. Please organize it in a better way to be easy to read.
Figure 4. Revised model. Please present the results of the study by numbers and significant paths in the presented figure.
The last sections (Discussion and implications) of the paper should be developed further to have better insights of the results, and should be reorganized as flowing:
6. Discussion and implications:
6.1 Discussion:
6.2 Theoretical implications
6.3 Practical implications
7. Limitations and future research directions
8. Conclusion
More up to date references should be added. Especially that no refences in 2023.
The paper should be carefully proofread for English language.
Comments on the Quality of English LanguageThe paper should be carefully proofread for English language.
Round 2
Reviewer 1 Report
Comments and Suggestions for Authors
In this round for my rising issues, I have no further comments.
Comments on the Quality of English LanguageIf possible, the authors should find a native English speaker help the editing.
Author Response
Thank you for taking the time to review this manuscript once again. We sincerely appreciate your valuable suggestions and find them extremely helpful.
Here is our response to your comments on the quality of English language:
Thank you for your suggestion on improving the English quality of our manuscript. We have taken your advice and sought the guidance of an English language expert for the manuscript. We found that some parts of the manuscript were indeed unclear and difficult to understand. Appropriate adjustments have been made to these sections. Please refer to the revised manuscript (Round 2).
Reviewer 3 Report
Comments and Suggestions for Authors
Thank you very much for giving me the chance to review the manuscript for the second time. I believe that the authors have addressed all the suggestions and comments. However, I have one more minor comment that should be included to develop the quality of the paper and to be more suitable for publication, as following:
· The results should be presented in numbers in Figure 4. Revised model (page 16), not the hypotheses as shown in the Figure.
Good luck with your publication, I wish you all the best.
Author Response
Thank you for taking the time to review this manuscript once again. We sincerely appreciate your valuable suggestions and find them extremely helpful.
Here is our response to your comment:
Comment: The results should be presented in numbers in Figure 4. Revised model (page 16), not the hypotheses as shown in the Figure.
Response: Thank you for your suggestion. We have optimized the Figure 4. Revised model based on the results of the path analysis (Table 5, page 12). Please refer to the revised manuscript (Round 2) page 15 for the details or the updated figure below.